# NEURAL TOPIC MODELING WITH LARGE LANGUAGE MODELS IN THE LOOP

## ABSTRACT

Topic modeling is a fundamental task in natural language processing, allowing the discovery of latent thematic structures in text corpora. While Large Language Models (LLMs) have demonstrated promising capabilities in topic discovery, their direct application to topic modeling suffers from issues such as incomplete topic coverage, misalignment of topics, and inefficiency. To address these limitations, we propose LLM-ITL, a novel LLM-in-the-loop framework that integrates LLMs with many existing Neural Topic Models (NTMs). In LLM-ITL, global topics and document representations are learned through the NTM, while an LLM refines the topics via a confidence-weighted Optimal Transport (OT)-based alignment objective. This process enhances the interpretability and coherence of the learned topics, while maintaining the efficiency of NTMs. Extensive experiments demonstrate that LLM-ITL can help NTMs significantly improve their topic interpretability while maintaining the quality of document representation.

## 1 INTRODUCTION

Topic modeling is an essential task in natural language processing that uncovers hidden thematic structures within large text collections in an unsupervised way. The ability to automatically extract topics has proven to be invaluable across a range of disciplines, such as bioinformatics (Liu et al., 2016), marketing research (Reisenbichler & Reutterer, 2019), and information retrieval (Yi & Allan, 2009). Topic models are usually based on probabilistic frameworks that generate a set of interpretable global topics, each represented as a distribution over vocabulary terms. These topics are then used to represent individual documents as mixtures of topics, providing a structured and interpretable view of the corpus. Recently, research on topic modeling has shifted from classical Bayesian methods such as Latent Dirichlet Allocation (LDA) (Blei et al., 2003) to Neural Topic Models (NTMs) (Zhao et al., 2021; Churchill & Singh, 2022; Wu et al., 2024) that use deep neural networks to model document-topic distributions, enabling more expressive and flexible representations compared to their probabilistic counterparts.

While Large Language Models (LLMs) (OpenAI, 2022; Touvron et al., 2023a;b) have redefined the landscape of natural language processing, topic models continue to hold their place as valuable tools for text analysis. Specifically, LLMs can provide a fine-grained understanding of a document; however, given a large collection of domain-specific documents, topic models are more suitable to obtain a clear global view of the topics in a more interpretable way with much less computational cost. Unsurprisingly, it has been a trending research direction to use LLMs to improve topic modeling (Rijcken et al., 2023; Wang et al., 2023; Pham et al., 2023; Mu et al., 2024; Doi et al., 2024; Chang et al., 2024). Despite the promising performance of these initial studies, most existing methods involve prompting LLMs to generate topics for each document in the corpus, which may lead to several limitations. As LLMs are asked to focus on a document individually, they may be unable to cover all the topics across all the documents in the corpus (Doi et al., 2024), which is critical in topic modeling. Moreover, although LLMs excel at capturing local context, they usually struggle with long documents with multiple interrelated topics, which may evolve or shift throughout the text. With their limited window of focus, LLMs may miss key topics of a document that are necessary to fully understand its content. Finally, it is computationally expensive as LLMs have to do inference for documents in the corpus; thus, existing methods usually scale poorly with large datasets.

To overcome the aforementioned limitations, we propose **LLM-ITL**, a framework that integrates LLMs into NTMs and enhances the overall quality and interpretability of the learned topics, while maintaining computational efficiency. Specifically, to enhance the interpretability of the topics learned by an NTM, we introduce an LLM-based refinement step. The representative words for each topic, as generated by the NTM, are provided to the LLM, which suggests improved words that better capture the semantic meaning of the topic. The refinement process is guided by a novel plug-in objective based on Optimal Transport (OT), which ensures that the topics learned by the NTM align closely with the LLM's refinements. Additionally, to mitigate potential hallucinations from the LLM (i.e., the generation of inaccurate or irrelevant suggestions), we introduce a confidence-weighted mechanism that adjusts the influence of the LLM's suggestions based on their confidence scores. Our proposed LLM-ITL framework offers the following key contributions:

- **Improved balance between topic coherence and document representation quality**: With the LLM's refined topics and OT-based alignment, the topics generated are more interpretable and semantically coherent. At the same time, LLM-ITL ensures that the document-topic distributions, as learned by the NTM, remain high-quality and reflective of the document's content.

- **Efficiency and scalability**: Unlike most existing LLM-based approaches that rely on document-level LLM analysis, LLM-ITL uses LLMs at the word level, significantly reducing computational overhead for large datasets.

- **Flexibility**: LLM-ITL is a modular framework that can integrate with a variety of NTMs and LLMs, offering flexibility in model selection depending on the application and computational constraints.

- **State-of-the-art performance**: Extensive experimental results on multiple benchmark datasets show that LLM-ITL achieves state-of-the-art performance in both topic coherence and document representation quality.

## 2 BACKGROUND

### 2.1 PROBLEM SETUP FOR TOPIC MODELING

Topic models have been popular across various fields for their ability to interpret text corpora in an unsupervised manner. Given a document collection $\mathcal{D} := \{\boldsymbol{d}_1, \ldots, \boldsymbol{d}_N\}$, a topic model learns to discover a set of global topics $\mathcal{T} := \{\boldsymbol{t}_1, \ldots, \boldsymbol{t}_K\}$, each of which is a distribution over the $V$ vocabulary words $\boldsymbol{t} \in \Delta^V$ ($\Delta$ denotes the probability simplex). Ideally, each topic represents a semantic concept that can be interpreted with its top-weighted words. At the document level, the topic model represents each document as a distribution over the $K$ topics, i.e., $\boldsymbol{z} \in \Delta^K$, which indicates the topic proportion of each topic within the document. The interpretability of topic models derives from both the corpus-level topics $\mathcal{T}$, and the document-level topical representation $\boldsymbol{z}$ for each document.

### 2.2 NEURAL TOPIC MODELS

A Neural Topic Model (NTM) (Miao et al., 2017; Srivastava & Sutton, 2017; Card et al., 2017; Dieng et al., 2020; Zhao et al., 2020; Xu et al., 2023a;b) is typically trained by modeling $p(\boldsymbol{z}|\boldsymbol{x})$ and $p(\boldsymbol{x}|\boldsymbol{z})$, where $\boldsymbol{x} \in \mathbb{N}^V$ represents the Bag-of-Words (BOWs) of a document. NTMs, which employ deep neural networks for topic modeling, are commonly based on Variational Auto-Encoders (VAEs) (Kingma & Welling, 2013) and Amortized Variational Inference (AVI) (Rezende et al., 2014). For VAE-NTMs, $p(\boldsymbol{x}|\boldsymbol{z})$ is modeled by a decoder network $\boldsymbol{\phi}$, i.e., $\boldsymbol{x} := \boldsymbol{\phi}(\boldsymbol{z})$. The posterior $p(\boldsymbol{z}|\boldsymbol{x})$ is approximated by $q(\boldsymbol{z}|\boldsymbol{x})$, which is modeled by an encoder network $\boldsymbol{\theta}$, i.e., $\boldsymbol{z} := \boldsymbol{\theta}(\boldsymbol{x})$. The training objective of VAE-NTMs is to maximize the Evidence Lower Bound (ELBO):

$$\max_{\boldsymbol{\theta}, \boldsymbol{\phi}} (\mathbb{E}_{q_{\boldsymbol{\theta}}(\boldsymbol{z}|\boldsymbol{x})}[\log p_{\boldsymbol{\phi}}(\boldsymbol{x}|\boldsymbol{z})] - \mathbb{KL}[q_{\boldsymbol{\theta}}(\boldsymbol{z}|\boldsymbol{x}) \parallel p(\boldsymbol{z})]), \tag{1}$$

where the first term encourages the reconstruction of the document, and the second is the Kullback–Leibler divergence between the approximate posterior and the prior distribution. By implementing a single linear layer for the decoder $\boldsymbol{\phi} \in \mathbb{R}^{V \times K}$, the $k$-th topic distribution $\boldsymbol{t}_k$ can be

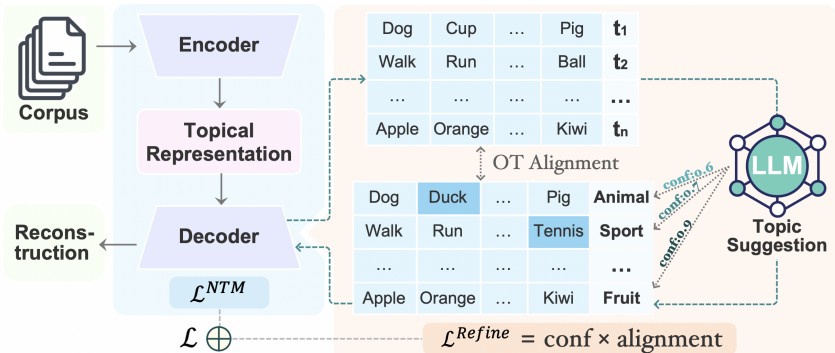

Figure 1: Overview of LLM-ITL. The topics and document representations are learned by the Neural Topic Model (NTM) component. Concurrently, a Large Language Model (LLM) suggests better topic words for the learned topics (e.g., decoder) from the NTM. An Optimal Transport (OT)-based topic alignment objective is proposed to align the word distribution between the topics from the NTM and those suggested by the LLM. The alignment is further weighted by the confidence of the LLM in providing the suggestions. This confidence-weighted topic refinement objective is plugged into the standard training of an NTM as the overall objective of LLM-ITL.

obtained by normalizing the $k$-th column of the decoder's weight matrix:

$$t_k := \text{softmax}(\phi_{:,k})^{\text{T}}. \tag{2}$$

Its topic words $w_k$ are obtained by taking the top-weight words of $t_k$, written as:

$$w_k := \mathcal{V}[f_{\text{topn}}(t_k, N)], \tag{3}$$

where $f_{\text{topn}}(a, N)$ defines a function that returns the indices of the top-$N$ values of vector $a$; $\mathcal{V}$ denotes the vocabulary set of the corpus.

## 2.3 OPTIMAL TRANSPORT

Optimal Transport (OT) has been widely used for comparing probability distributions (Cuturi, 2013; Frogner et al., 2015; Seguy et al., 2017; Peyré et al., 2019). Let $\mu(x, a) := \sum_{i=1}^{N} a_i \delta_{x_i}$ and $\mu(y, b) := \sum_{j=1}^{M} b_j \delta_{y_j}$ be two discrete distributions, where $a := [a_1, \ldots, a_N]$ and $b := [b_1, \ldots, b_M]$ are the probability vectors; $x := \{x_1, \ldots, x_N\}$ and $y := \{y_1, \ldots, y_M\}$ are the supports of these two distributions. The OT distance between $\mu(x, a)$ and $\mu(y, b)$ is obtained by finding the optimal transport plan $P^*$ that transports the probability mass from $a \in \Delta^N$ to $b \in \Delta^M$, written as following:

$$d_{\text{OT}}(\mu(x, a), \mu(y, b)) := \min_{P} \sum_{i=1}^{N} \sum_{j=1}^{M} C_{i,j} P_{i,j}, \tag{4}$$

subject to $\sum_{j=1}^{M} P_{i,j} = a_i, \forall i = 1, \ldots, N$ and $\sum_{i=1}^{N} P_{i,j} = b_j, \forall j = 1, \ldots, M$. Here, $P \in \mathbb{R}_{\geq 0}^{N \times M}$ is the transport plan, with entry $P_{i,j}$ indicating the amount of probability mass moving from $a_i$ to $b_j$; $C \in \mathbb{R}_{\geq 0}^{N \times M}$ denotes the cost matrix, with entry $C_{i,j}$ specifying the distance between supports $x_i$ and $y_j$. Various OT solvers (Flamary et al., 2021) have been proposed to compute the OT distance.

## 3 METHOD

In this work, we propose LLM-ITL, an LLM-in-the-loop framework that efficiently integrates the LLM with the training of NTMs, offering a more interpretable and comprehensive topic modeling pipeline. An overview of LLM-ITL is illustrated in Figure 1. LLM-ITL involves the following key components: LLM-based topic suggestion, OT distance for topic alignment, and confidence-weighted topic refinement.

## 3.1 LLM-BASED TOPIC SUGGESTION

During the training of an NTM, it typically generates a set of topics, where each topic is represented by a distribution over words, with the highest-probability words forming the core "meaning" of the topic. While these words offer a rough semantic grouping, they often lack clarity or precision, leading to difficulties in interpretation. For instance, topics may contain words that are too general, too specific, or semantically ambiguous, making it hard for users to derive clear labels or understand the thematic focus of the topic.

To address this, LLM-ITL proposes to use LLMs to suggest better words or labels that more clearly express the same underlying concept. The LLM is prompted with the top words from each topic, and it generates two outputs: a topic label, which is a concise and interpretable summary of the topic; and a set of refined topic words, which better represent the underlying semantic concept of the topic. This process capitalizes on the LLM's ability to grasp language nuances and provide more semantically rich suggestions for the topic. The LLM's extensive pre-training on diverse and large datasets allows it to capture subtle relationships between words that may not be apparent in the purely statistical or neural-based methods employed by NTMs.

To obtain the topic label and refined words in a structured manner, *chain-of-thought (CoT) prompting* (Wei et al., 2022) is employed. CoT prompting encourages the LLM to reason step-by-step through the task, ensuring that it carefully considers the topic words before generating a label and refinement. The LLM's output sequence $s$ includes both the *topic label* and *refined words*, extracted as follows for each set of topic words:

$$s := \theta^{\mathrm{llm}}(\mathrm{Prompt}(\boldsymbol{w})),$$

$$\text{Topic label } \boldsymbol{w}^l : (s_{\text{start of label}}, \dots, s_{\text{end of label}}),$$

$$\text{Refined words } \boldsymbol{w}' : (s_{\text{start of words}}, \dots, s_{\text{end of words}}), \tag{5}$$

where $\boldsymbol{w}$ represents the original topic words; $\theta^{\mathrm{llm}}$ denotes the LLM model; the topic label $\boldsymbol{w}^l$ and the refined words $\boldsymbol{w}'$ are extracted as subsequences from the LLM's output $s$. The used prompt is illustrated in Appendix A.1. A study of prompt variants is provided in Appendix H.

## 3.2 OT-BASED TOPIC ALIGNMENT

A key innovation in LLM-ITL is the use of Optimal Transport (OT) distance to align the topic word distributions generated by the NTM with the refined topic word distributions provided by the LLM. OT is a mathematical framework that computes the "cost" of transforming one probability distribution into another, making it an ideal tool for measuring the alignment between two sets of words (Kusner et al., 2015; Yang et al., 2024).

Formally, given a set of original topic words $\boldsymbol{w} := \{w_1, w_2, \dots, w_N\}$ with probability vector $\boldsymbol{t} := [t_1, t_2, \dots, t_N]$ as obtained by Eq. 3 and Eq. 2, respectively; as well as refined topic words $\boldsymbol{w}' := \{w'_1, w'_2, \dots, w'_M\}$ with probability vector $\boldsymbol{u} := [u_1, u_2, \dots, u_M]^1$ from the LLM, the OT distance between these two word distributions can be formulated as:

$$d_{\mathrm{OT}}(\mu(\boldsymbol{w}, \boldsymbol{t}), \mu(\boldsymbol{w}', \boldsymbol{u})) = \min_{\boldsymbol{P}} \sum_{i=1}^{N} \sum_{j=1}^{M} C_{i,j} P_{i,j}, \tag{6}$$

where $\boldsymbol{P} \in \mathbb{R}_{\geq 0}^{N \times M}$ is the transport plan, with entry $P_{i,j}$ denoting the amount of probability mass transported from $t_i$ to $u_j$; $\boldsymbol{C} \in \mathbb{R}_{\geq 0}^{N \times M}$ is the cost matrix, where $C_{i,j}$ represents the cost of transporting mass between word $w_i$ and $w'_j$.

The cost matrix $\boldsymbol{C}$ is constructed using the cosine distance between pre-trained word embeddings $\mathcal{E}^{\boldsymbol{w}} := \{\boldsymbol{e}^{w_1}, \boldsymbol{e}^{w_2}, \dots, \boldsymbol{e}^{w_N}\}$ (for the original topic words) and $\mathcal{E}^{\boldsymbol{w}'} := \{\boldsymbol{e}^{w'_1}, \boldsymbol{e}^{w'_2}, \dots, \boldsymbol{e}^{w'_M}\}$ (for the refined topic words). The cosine distance for each entry $C_{i,j}$ is computed as:

$$C_{i,j} := d_{\cos}(\boldsymbol{e}^{w_i}, \boldsymbol{e}^{w'_j}), \tag{7}$$

---

[1]We assume each of the refined topic words from the LLM is equally important, thus $\boldsymbol{u}$ is a uniform probability vector.

where $d_{\cos}(\boldsymbol{a}, \boldsymbol{b})$ denotes the cosine distance between the embedding vectors $\boldsymbol{a}$ and $\boldsymbol{b}$.

By minimizing this OT distance, the learned topic words from the NTM become aligned with the refined words suggested by the LLM, leading to more semantically coherent topics. This OT-based refinement loss is incorporated into the overall training objective, guiding the NTM to adjust its learned topics to match the LLM's refined representations.

### 3.3 CONFIDENCE-WEIGHTED TOPIC REFINEMENT

LLMs, despite their powerful language capabilities, can sometimes produce hallucinated outputs—irrelevant or incorrect suggestions that do not align with the input data (Ji et al., 2023). To mitigate the impact of such hallucinations, LLM-ITL introduces a confidence-weighted refinement mechanism. The confidence mechanism assesses the reliability of the LLM's refinements and adjusts their influence on the NTM's training accordingly. This ensures that high-confidence refinements have a greater impact on the final topic representation, and vice versa.

We propose two methods for calculating topic labeling confidence, considering whether the LLM is open-source or not: (1) Label token probability, applicable for open-source LLMs where the token probability of their generation is accessible; (2) Word intrusion confidence, available for both open and closed-source LLMs.

**Label Token Probability**  This method computes the product of the token probabilities for the topic label generated by the LLM. It reflects the LLM's certainty in generating the specific topic label:

$$\text{Conf}(\boldsymbol{w}^l)^{\text{prob}} := \prod_{i=sol}^{eol} p(s_i | \boldsymbol{s}_{<i}, \boldsymbol{c}), \tag{8}$$

where "$sol$" and "$eol$" denote the indices of "start of label" and "end of label" token, respectively; $p(s_i | \boldsymbol{s}_{<i}, \boldsymbol{c})$ denotes the token probability of the $i$-th token; $\boldsymbol{c}$ denotes the input context to the LLM.

**Word Intrusion Confidence**  This method evaluates the proportion of irrelevant or "intruder" words removed by the LLM during suggestion. A topic label generated based on a higher rate of intruder removal indicates that it is harder for the LLM to identify the topic from the original topic words, leading to lower confidence:

$$\text{Conf}(\boldsymbol{w}^l)^{\text{intrusion}} := 1 - \frac{N^{\text{intruder}}}{N^{\boldsymbol{w}}}, \tag{9}$$

where $N^{\boldsymbol{w}}$ denotes the number of words in the given topic; $N^{\text{intruder}}$ denotes the number of intruders identified by the LLM.

By incorporating the topic labeling confidence as a weight for the topic alignment loss, we adaptively adjust the impact of the LLM's suggestion based on the confidence score. We write our confidence-weighted topic refinement objective as follows:

$$\min_{\boldsymbol{\phi}} \sum_{k=1}^{K} \text{Conf}(\boldsymbol{w}_k^l) \, d_{\text{OT}}(\mu(\boldsymbol{w}_k, \boldsymbol{t}_k), \mu(\boldsymbol{w}_k', \boldsymbol{u}_k)). \tag{10}$$

### 3.4 INTEGRATION WITH NTMs

One of the core strengths of LLM-ITL lies in its flexibility to integrate with various NTMs while leveraging the semantic capabilities of LLMs for topic refinement. The framework is designed to complement and enhance NTMs, providing an efficient and interpretable topic modeling pipeline. The LLM-ITL framework is highly modular and can be seamlessly integrated with a wide range of NTMs. Here, we focus on the VAE-NTM framework that accommodate many NTMs (Miao et al., 2017; Srivastava & Sutton, 2017; Card et al., 2017; Dieng et al., 2020; Zhao et al., 2020; Nguyen & Luu, 2021; Xu et al., 2023a;b), while our framework is not limited to VAE-NTMs only. By integrating the topic refinement objective with the training of an NTM, we obtain the overall objective of LLM-ITL:

$$\min_{\boldsymbol{\Theta}} (\mathcal{L}^{\text{ntm}} + \gamma \cdot \mathbf{I}(t > T^{\text{refine}}) \cdot \mathcal{L}^{\text{refine}}), \tag{11}$$

where $\boldsymbol{\Theta} := \{\boldsymbol{\theta}, \boldsymbol{\phi}\}$ denotes model parameters; $\mathcal{L}^{\text{ntm}}$ and $\mathcal{L}^{\text{refine}}$ denote the NTM loss and refinement loss in Eq. 1 and Eq. 10, respectively; $\gamma$ controls the strength of focusing on the LLM's refinements; $t$ and $T^{\text{refine}}$ denote the current training step and the start of topic refinement, respectively; $\mathbf{I}(\cdot)$ denotes the indicator function, which ensures that the refinement process only starts after the NTM has learned a stable topic representation, allowing the model to capture the core structure of the corpus before fine-tuning the topics with LLM guidance. The algorithm of LLM-ITL is provided in Appendix B.

## 4 Related Work

**Topic Models**   Classical topic models, such as Latent Dirichlet Allocation (LDA) (Blei et al., 2003) and its variants (Blei & Lafferty, 2006; Rosen-Zvi et al., 2012; Yan et al., 2013), are Bayesian probabilistic models with various generative assumptions about the documents. Neural Topic Models (NTMs) (Miao et al., 2017; Srivastava & Sutton, 2017; Card et al., 2017; Dieng et al., 2020; Zhao et al., 2020; Nguyen & Luu, 2021; Xu et al., 2023a;b) use deep neural networks to learn topics and document representations, and are commonly based on Variational Autoencoders (VAE) (Kingma & Welling, 2013) and Amortized Variational Inference (AVI) (Rezende et al., 2014). Clustering-based topic models (Sia et al., 2020; Grootendorst, 2022) discover topics using clustering algorithms based on embeddings from pre-trained language models. Ultimately, the capability of these models to interpret a corpus is limited by the top words representation of each topic.

**LLMs in Topic Modeling**   LLMs have been involved in topic modeling in various ways. Rijcken et al. (2023) investigate the use of ChatGPT (OpenAI, 2022) to generate descriptions for topic words and found the effectiveness of these topic descriptions. Recent works leverage LLMs for topic model evaluation in different ways, such as applying LLMs for word intrusion or topic rating for topics (Rahimi et al., 2023; Stammbach et al., 2023), or keyword generation for documents (Yang et al., 2024). LLM-based topic models have emerged (Wang et al., 2023; Pham et al., 2023; Mu et al., 2024; Doi et al., 2024), which prompt LLMs to generate topics and assign topics to documents. Different from these methods that focus on the document-level, ours prompts LLMs to suggest better topic words which are used to refine the training of NTMs. More recently, Chang et al. (2024) show that LLMs are effective at refining topic words, leading to improved topic coherence. However, their method refines the topic words of trained topic models in a post-hoc manner, while ours is a regularization term for training NTMs. Our method is also loosely related to uncertainty estimation of LLMs and we omit the discussion on this in Appendix C.

## 5 Experiments

### 5.1 Experimental Setup

**Datasets**   We conduct experiments on four widely used datasets in topic modeling, including 20Newsgroup (Lang, 1995) (**20News**), Reuters-21578 (Aletras & Stevenson, 2013) (**R8**), **DBpedia** (Auer et al., 2007) and **AGNews** (Zhang et al., 2015). Further details of these datasets are described in the Appendix D.1. The number of mined topics (i.e., $K$) is commonly regarded as a hyper-parameter for the dataset (Zhao et al., 2020; Wu et al., 2024). For datasets containing long documents, such as 20News and R8, we set the number of topics to 50. For datasets with short documents, such as DBpedia and AGNews, we set the number to 25. We also run experiments at different $K$ values, which are reported in Appendix E.

**Baselines**   We compare LLM-ITL with topic models of different types, including Latent Dirichlet Allocation (**LDA**) (Blei et al., 2003); Neural Variational Document Model (**NVDM**) (Miao et al., 2017); LDA with Products of Experts (**PLDA**) (Srivastava & Sutton, 2017); Embedded Topic Model (**ETM**) (Dieng et al., 2020); Neural Topic Model with Covariates, Supervision, and Sparsity (**SCHOLAR**) (Card et al., 2017); Contrastive Learning Neural Topic Model (**CLNTM**) (Nguyen & Luu, 2021); **BERTopic** (Grootendorst, 2022) and **TopicGPT** (Pham et al., 2023). Further details about these baselines and their settings are provided in Appendix D.2.

Table 1: Topic coherence (NPMI) and topic alignment (PN). The best and second-best performance of each column are highlighted in boldface and underlined, respectively. "NA" indicates the evaluation is not applicable. The performance improvement of LLM-ITL over its base model is computed.

| Model | 20News | | R8 | | DBpedia | | AGNews | |
|---|---|---|---|---|---|---|---|---|
| | NPMI | PN | NPMI | PN | NPMI | PN | NPMI | PN |
| LDA (Blei et al., 2003) | 3.95 ± 0.27 | 0.489 ± 0.009 | -3.43 ± 0.49 | **0.700** ± 0.005 | 6.42 ± 0.22 | 0.762 ± 0.013 | 7.74 ± 0.44 | 0.596 ± 0.008 |
| NVDM (Miao et al., 2017) | -16.89 ± 0.71 | 0.145 ± 0.006 | -8.12 ± 0.62 | 0.360 ± 0.012 | -7.82 ± 0.49 | 0.185 ± 0.004 | -7.39 ± 0.85 | 0.248 ± 0.013 |
| PLDA (Srivastava & Sutton, 2017) | -13.70 ± 0.76 | 0.101 ± 0.005 | -6.26 ± 0.45 | 0.524 ± 0.009 | 5.60 ± 0.58 | 0.653 ± 0.008 | 7.03 ± 0.77 | 0.487 ± 0.011 |
| BERTopic (Grootendorst, 2022) | 2.18 ± 0.73 | 0.342 ± 0.008 | -0.65 ± 0.14 | 0.697 ± 0.001 | 7.68 ± 0.84 | 0.720 ± 0.009 | 5.00 ± 0.92 | 0.450 ± 0.010 |
| TopicGPT (Pham et al., 2023) | NA | 0.363 ± 0.000 | NA | 0.410 ± 0.000 | NA | 0.706 ± 0.000 | NA | 0.634 ± 0.000 |
| ETM (Dieng et al., 2020) | 2.96 ± 0.42 | 0.404 ± 0.010 | -1.71 ± 0.72 | 0.669 ± 0.009 | 4.49 ± 0.45 | 0.762 ± 0.011 | 6.15 ± 0.59 | 0.568 ± 0.008 |
| LLM-ITL (ETM) | **8.92** ± 0.74 | 0.398 ± 0.010 | **7.13** ± 0.57 | 0.686 ± 0.012 | 14.83 ± 0.79 | 0.742 ± 0.016 | **12.04** ± 0.95 | 0.569 ± 0.005 |
| | ↑ 5.96 | ↓ 0.006 | ↑ 8.84 | ↑ 0.017 | ↑ 10.34 | ↓ 0.020 | ↑ 5.89 | ↑ 0.001 |
| SCHOLAR (Card et al., 2017) | -2.12 ± 0.77 | **0.582** ± 0.010 | -4.06 ± 0.15 | 0.680 ± 0.013 | 12.32 ± 1.54 | 0.825 ± 0.015 | 6.41 ± 0.70 | 0.638 ± 0.003 |
| LLM-ITL (SCHOLAR) | 7.58 ± 0.45 | 0.568 ± 0.010 | -0.78 ± 0.60 | 0.680 ± 0.012 | **15.13** ± 1.61 | **0.828** ± 0.013 | 11.07 ± 0.78 | **0.639** ± 0.002 |
| | ↑ 9.70 | ↓ 0.014 | ↑ 3.28 | ↑ 0.000 | ↑ 2.81 | ↑ 0.003 | ↑ 4.66 | ↑ 0.001 |
| CLNTM (Nguyen & Luu, 2021) | -2.21 ± 1.07 | 0.575 ± 0.011 | -4.99 ± 0.36 | 0.691 ± 0.005 | 3.75 ± 1.35 | 0.683 ± 0.040 | 5.20 ± 1.38 | 0.607 ± 0.014 |
| LLM-ITL (CLNTM) | 8.12 ± 0.49 | 0.576 ± 0.005 | -1.25 ± 0.57 | 0.691 ± 0.005 | 10.03 ± 1.11 | 0.684 ± 0.039 | 11.13 ± 0.96 | 0.594 ± 0.011 |
| | ↑ 10.33 | ↑ 0.001 | ↑ 3.74 | ↑ 0.000 | ↑ 6.28 | ↑ 0.001 | ↑ 5.93 | ↓ 0.013 |

**Settings of LLM-ITL**    LLM-ITL is a framework compatible with most NTMs and LLMs. We use ETM, SCHOLAR, and CLNTM as the base models for our experiments. We use LLAMA3-8B-Instruct[2] in LLM-ITL for main experiments. For OT computation, we use GloVe (Pennington et al., 2014) word embeddings pre-trained on Wikipedia to construct the OT cost matrix, and compute the OT distance using the POT[3] package. For the topic labeling confidence, we use label token probability in Eq. 8 for our main experiments. As for the hyper-parameters of LLM-ITL, we set the topic refinement strength $\gamma$ to 200; and the refinement step $T^{\text{refine}}$ to 150 for ETM and 450 for SCHOLAR and CLNTM. We set the number of words for the topic label to 2 when prompting the LLM. All hyper-parameters of LLM-ITL are studied in the following sections. As for the LLM generation, we use greedy decoding to enable deterministic output and set the maximum new generation tokens to 300. Each trial[4] of LLM-ITL in our experiment takes a few hours on a single 80GB A100 GPU.

**Evaluation Metrics**    We evaluate both the topic quality and the document representation quality for topic models. For topic quality, we apply the widely used **topic coherence** metric, Normalized Pointwise Mutual Information (**NPMI**) (Lau et al., 2014). We report the average NPMI values (in percentage) of all topics. Moreover, topic diversity (Dieng et al., 2020) is also evaluated, which is reported in Appendix E.3. For document representation quality, we evaluate the alignment between a document's true label and the top-weighted topic of its topical representation using external clustering metrics, known as **topic alignment** (Chuang et al., 2013; Pham et al., 2023). We compute the commonly used Purity and Normalized Mutual Information (NMI) to evaluate clustering performance. Since Purity and NMI are considered equally important, fall within the same range (from 0 to 1), and are often reported together, we report their average as **PN**, serving as an overall indicator of topic alignment performance. Detailed results for Purity and NMI are provided in Appendix E.4. Intuitively, topic coherence (e.g., NPMI) reflects how coherent the learned topic words are, while topic alignment (e.g., PN) indicates how well the model represents the documents through the learned topics. Further details on the calculation of these metrics are provided in Appendix D.3.

## 5.2    RESULTS

**Topic Coherence and Alignment**    We show the performance of topic coherence and topic alignment for different models in Table 1. We summarize the following remarks based on the results: (1) As for topic coherence, LLM-ITL significantly improves the performance of the base model and achieves state-of-the-art (SOTA) performance. (2) In terms of topic alignment, LLM-ITL inherits the document representation capability of its base model and shows SOTA performance in most cases. (3) Moreover, for long-document corpora such as 20News and R8, LLM-ITL outperforms existing LLM-based topic models like TopicGPT in terms of topic alignment, where the LLM alone

---

[2]https://huggingface.co/meta-llama/Meta-Llama-3-8B-Instruct

[3]https://pythonot.github.io/

[4]All experiments are conducted five times with different model random seeds throughout this work. The mean and standard deviation values of performance are reported.

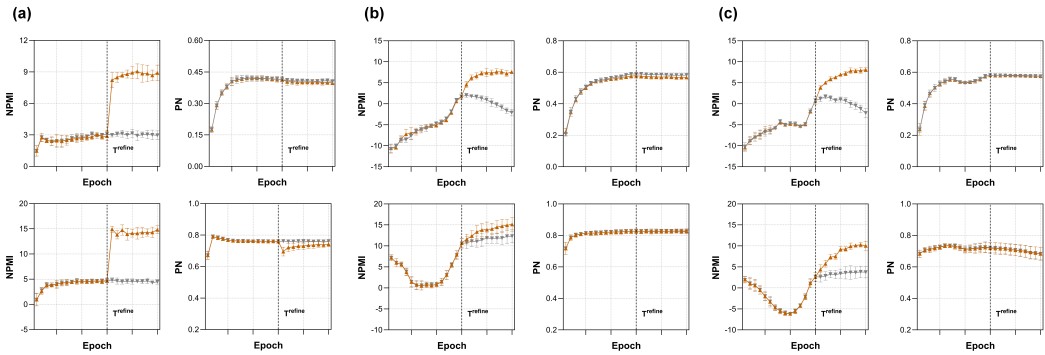

Figure 2: Learning curves of LLM-ITL with different base models in terms of topic coherence (NPMI) and topic alignment (PN) on 20News (**Top**) and DBpedia (**Bottom**). The **gray** and **brown** curves indicate the status of the base model and LLM-ITL, respectively. The base models are (**a**) ETM, (**b**) SCHOLAR and (**c**) CLNTM, respectively.

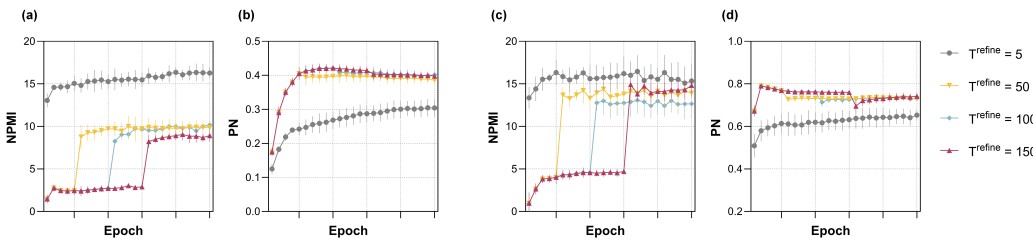

Figure 3: Learning curves of LLM-ITL (ETM) with different $T^{\text{refine}}$ in terms of topic coherence (NPMI) and topic alignment (PN) on 20News ((**a**) and (**b**)) and DBpedia ((**c**) and (**d**)).

may fail to fully capture the topics for long documents. Additional results about topic coherence (Appendix E.1) and alignment (Appendix E.2) at different settings of $K$, as well as topic diversity (Appendix E.3) performance, are illustrated in the appendix.

**Learning Status of LLM-ITL**    To clearly demonstrate how LLM-ITL improves its base model, we illustrate the learning curves for topic coherence and alignment on both a long document dataset (e.g., 20News) and a short document dataset (e.g., DBpedia) in Figure 2. We have the following observations based on the results: (1) When topic refinement is applied, LLM-ITL significantly improves the topic coherence [5] of the base model. (2) LLM-ITL has little overall influence on topic alignment in most cases.

**Balancing Topic Coherence and Alignment**    As indicated by previous works (Bhatia et al., 2017; Yang et al., 2024), a topic model with better topic coherence may not perform well in document representations (i.e., topic alignment) at the same time, and vice versa. To provide further insights into how LLM-ITL balances between topic coherence and alignment, we illustrate the learning curves of LLM-ITL (ETM) with different $T^{\text{refine}}$ in terms of both metrics. As illustrated in Figure 3, we have the following observations: (1) While starting topic refinement earlier (e.g., $T^{\text{refine}} = 5$) can lead to greater improvements in topic coherence, it may also introduce more irrelevant information about the corpus that is from the LLM's knowledge, thereby harming topic alignment performance. (2) For larger values of $T^{\text{refine}}$, the performance in terms of both topic coherence and alignment is comparable, indicating little sensitivity to the settings of $T^{\text{refine}}$ in a certain range. These observations

---

[5]Topic coherence is not correlated with the training of topic models (Chang et al., 2009), which can result in a drop in coherence in the learning curve.

Table 2: Examples of different topic models' output for a given document from 20News. Only the document's top assigned/weighted ($>= 0.1$) topics of its topical proportion/representation are listed in LDA and LLM-ITL.

| Document | Model | Topic | Proportion |
|---|---|---|---|
| Here's a listing that I came accross a while ago. This question seems to come up often enough that I figured this would be of interest. Note that the server "X Appeal" for DOS is available in demo form on the internet via anonymous ftp. This is one way of quickly checking out the feasability of using your system as an X server. Enjoy! - Pete *** Many words omitted here *** 1280x960x1 (TT, SM194) color 320x200x4 color 640x200x2 color 640x480x4 color 320x480x8 Ethernet Card: Atari Card (Mega or VME bus) Riebl/Wacker (Mega or VME bus) —— End Enclosure —— | LDA | [1] window file program problem run work running machine time version | 0.23 |
| | | [2] card driver monitor color video mode vga window screen problem | 0.19 |
| | | [3] image data graphic package software program format tool file processing | 0.16 |
| | | [4] mb mac mhz bit chip card scsi ram cpu memory | 0.10 |
| | | ... | ... |
| | TopicGPT | [1] Software Development – The document provides a list of X servers that can be used on non-UNIX networked machines. | NA |
| | | [2] Internet Culture – The document mentions the availability of X servers for various operating systems and provides information on how to access the file via anonymous ftp. | NA |
| | LLM-ITL | [1] Software Development – application model version program software designed tool development implementation window | 0.23 |
| | | [2] Computer Hardware – computer bios disk pc chip controller ram memory card apple | 0.13 |
| | | [3] Service Support – support provide offer access available providing includes provides use feature | 0.11 |
| | | ... | ... |

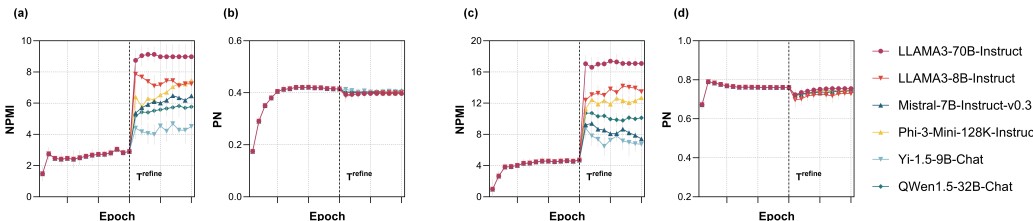

Figure 4: Learning curves of LLM-ITL (ETM) with different LLMs in terms of topic coherence (NPMI) and topic alignment (PN) on 20News ((**a**) and (**b**)) and DBpedia ((**c**) and (**d**)).

suggest the effectiveness of $T^{\text{refine}}$ in balancing topic coherence and alignment in LLM-ITL. For more hyper-parameter studies of LLM-ITL, see Appendix G.

**Qualitative Analysis** During the inference phase, LLM-ITL infers the topic proportion (i.e., topical representation) for a given document from the NTM component, and obtains the topic label from the LLM component, as shown in Table 2 (where ETM is used as the base model). We can observe that (1) Compared to topic models with top-words topics such as LDA, LLM-ITL provides more coherent topic words and offers topic labels, making the semantic meaning of the topics easier to identify. (2) Compared to the LLM-based topic model TopicGPT, LLM-ITL can obtain topic proportions as an indicator of the importance or relevance of topics to the document, offering more practical usage. For example, for TopicGPT, "Internet Culture" should be less relevant for the example document than "Software Development" if a good topic proportion is available.

**Flexibility with different LLMs** LLM-ITL is a framework compatible with most LLMs. Here, we examine the flexibility of LLM-ITL by integrating it with various LLMs. Apart from LLAMA3-8B-Instruct[6] (Dubey et al., 2024), we implement LLM-ITL with the latest open-sourced LLMs, including Mistral-7B-Instruct-v0.3[7] (Jiang et al., 2023), Phi-3-Mini-128K-Instruct[8] (Abdin et al., 2024), Yi-1.5-9B-Chat[9] (Young et al., 2024), Qwen1.5-32B-Chat[10] (Bai et al., 2023) and LLAMA3-70B-Instruct[11] (Dubey et al., 2024). As shown in Figure 4, LLM-ITL consistently improves topic

---

[6]https://huggingface.co/meta-llama/Meta-Llama-3-8B-Instruct

[7]https://huggingface.co/mistralai/Mistral-7B-Instruct-v0.3

[8]https://huggingface.co/microsoft/Phi-3-mini-128k-instruct

[9]https://huggingface.co/01-ai/Yi-1.5-9B-Chat

[10]https://huggingface.co/Qwen/Qwen1.5-32B-Chat

[11]https://huggingface.co/meta-llama/Meta-Llama-3-70B-Instruct

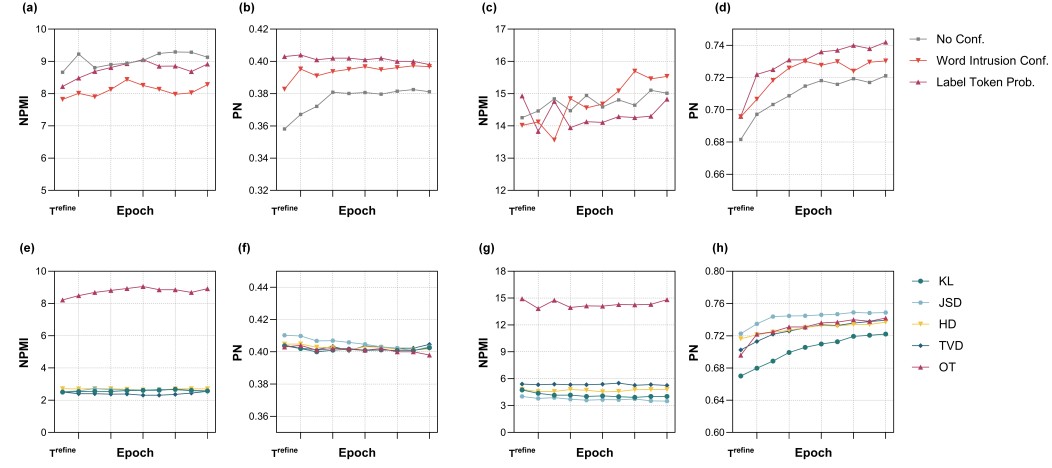

Figure 5: Ablation studies. **First row**: Ablation for confidence on 20News ((**a**) and (**b**)) and DBpedia ((**c**) and (**d**)); **Second row**: Ablation for OT on 20News ((**e**) and (**f**)) and DBpedia ((**g**) and (**h**)). Error bars are omitted for clarity in the figure.

coherence of its base model across different LLMs, and the improvement can be further enhanced when using larger LLMs such as LLAMA3-70B-Instruct, demonstrating the flexibility of LLM-ITL.

**Ablation Study for Confidence**   Here, we investigate the effectiveness of including the confidence scores during topic refinement. We apply **No Conf.** (i.e., $\text{Conf}(\boldsymbol{w}^l) = 1$ in Eq. 10 for all refinement), **Label Token Prob.** (i.e., Eq. 8) and **Word Intrusion Conf.** (i.e., Eq. 9) to LLM-ITL (ETM). We plot the learning curves for both metrics on 20News and DBpedia. From the results in Figure 5 (first row), we can observe that label token probability and word intrusion confidence consistently yield better performance in terms of PN. This suggests that by including proposed confidence during topic refinement, we reduce potential noisy suggestions from the LLM and achieve better topical representation for documents. For further studies on alternative LLM confidence measures, see Appendix F.

**Ablation Study for OT**   Here, we study the effectiveness of our OT-based topic refinement. We apply different metrics to measure the difference between the topic word distributions from the NTM and those from the LLM, including Kullback–Leibler (**KL**) divergence, Jensen–Shannon Divergence (**JSD**), Hellinger Distance (**HD**), and Total Variation Distance (**TVD**). As illustrated in Figure 5 (second row), our OT-based approach significantly benefits topic coherence compared to other distribution measurements.

## 6 CONCLUSION

In this paper, we introduced LLM-ITL, a novel framework that integrates Large Language Models (LLMs) with Neural Topic Models (NTMs) to address the limitations of both traditional topic models and the direct use of LLMs for topic discovery. By incorporating a confidence-weighted Optimal Transport (OT)-based topic alignment, LLM-ITL improves the interpretability and coherence of topics while maintaining the quality of document representations. Our framework effectively leverages the strengths of both LLMs and NTMs, offering a flexible, scalable, and efficient solution for topic modeling. Extensive experiments on benchmark datasets demonstrate that the LLM-ITL variants of NTMs achieve state-of-the-art performance in terms of topic coherence and document representation. In terms of limitations, the framework's reliance on LLM-generated refinements introduces a dependency on the quality of the LLM's outputs, which may vary based on the model used. Moreover, the method has been primarily evaluated on benchmark datasets, and its performance in more domain-specific corpora may require further investigation.

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

# A PROMPTS

## A.1 TOPIC SUGGESTION

**Prompt: Topic Suggestion with CoT**

You are a helpful assistant in understanding topics and words. Analyze step-by-step and provide the final answer.

Step 1. Given a set of words, summarize a topic (avoid using proper nouns as topics) by 2 words that covers most of those words. Note, only the topic, no other explanations.

Step 2. Remove irrelevant words about the topic from the given word list. Note, only the removed words, no other explanations.

Step 3. Add new relevant words (maximum 10 words) about the topic to the word list up to 10 words. Note, only the added words, no other explanations.

Step 4. Provide your answer in json format as {'Topic': '<2 Word Topic>', 'Words': '<Refined 10 Word List>'}. Note, only 1 0 refined words allowed for the topic, and no follow up explanations.

book, university, bank, science, vote, gordon, surrender, intellect, skepticism, shameful

**Output: Topic Suggestion with CoT**

Here is the analysis:

Step 1: Summarize the topic by 2 words that covers most of the words.
Topic: "Higher Learning"

Step 2: Remove irrelevant words about the topic from the given word list.
Removed words: "bank", "vote", "gordon", "surrender", "shameful"

Step 3: Add new relevant words (maximun 10 words) about the topic to the word list up to 10 words.
Added words: "education", "knowledge", "research", "curriculum", "degrees", "graduation", "professor", "thesis", "academic", "intellectual"

Step 4: Provide the answer in json format.
{"Topic": "Higher Learning",
"Words": ["book", "university", "science", "intellect", "education", "knowledge", "research", "curriculum", "degrees", "academic"]}

Figure A1: Prompt and output of topic suggestion with CoT. We take the product of token probabilities of topic label (e.g., words in **red** color) as the label token probability. We take the proportion of intruders (e.g., words in **green** color) as the word intrusion confidence.

## A.2 TOPIC SUGGESTION WITH VERBALIZED CONFIDENCE

**Prompt: Topic Suggestion with CoT and Verbalized Confidence**

You are a helpful assistant in understanding topics and words. Analyze step-by-step, provide the final answer and your confidence of solving the problem.

Step 1. Given a set of words, summarize a topic (avoid using proper nouns as topics) by 2 words that covers most of those words. Note, only the topic, no other explanations.

Step 2. Remove irrelevant words about the topic from the given word list. Note, only the removed words, no other explanations.

Step 3. Add new relevant words (Maximum 10 words) about the topic to the word list up to 10 words. Note, only the added words, no other explanations.

Step 4. Provide your answer in json format as {'Topic': '<2 Word Topic>', 'Words': '<Refined 10 Word List>', 'Confidence': '<Your confidence of solving the problem, numeric value from 0-100>'}. Note, only 10 refined words allowed for the topic, and no follow up explanations.

book, university, bank, science, vote, gordon, surrender, intellect, skepticism, shameful

**Output: Topic Suggestion with CoT and Verbalized Confidence**

Here is the analysis:

Step 1: Summarize the topic by 2 words that covers most of those words.
The 2-word topic is: "Higher Education"

Step 2: Remove irrelevant words about the topic from the given word list.
Removed words: "bank", "vote", "gordon", "surrender", "shameful"

Step 3: Add new relevant words (maximun 10 words) about the topic to the word list up to 10 words.
Added words: "research", "knowledge", "degrees", "campus", "students", "professors", "curriculum", "thesis", "graduation", "diploma"

Step 4: Provide the answer in json format.
{'Topic': 'Higher Education',
'Words': ['book', 'university','science', 'intellect', 'skepticism', 'research', 'knowledge', 'degrees', 'campus', 'students'],
'Confidence': 90}

Figure A2: Prompt and output of topic suggestion with CoT and verbalized confidence. The number in **red** color in the LLM's output represents the verbalized confidence.

## A.3 SELF-REFLECT CONFIDENCE

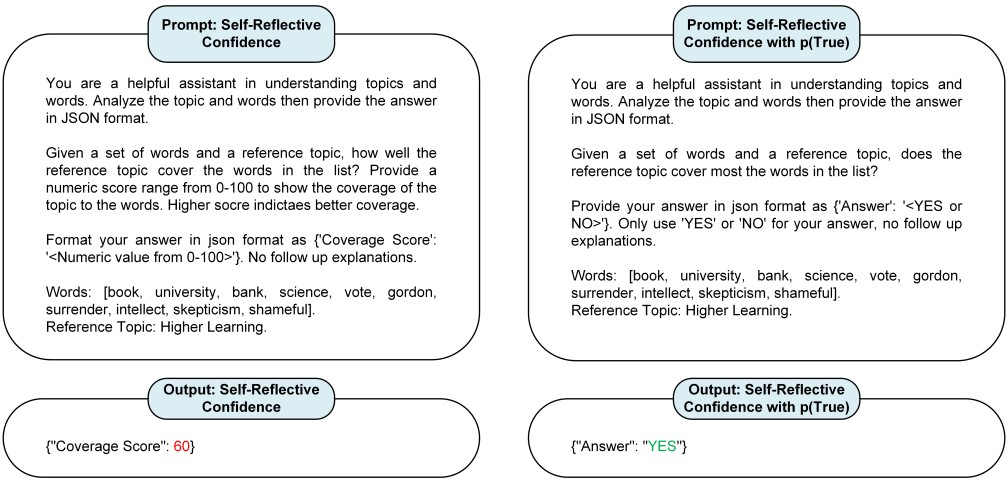

Figure A3: Prompt and output of self-reflective confidence and p(True). **Left**: self-reflective confidence where the number in **red** color represents the confidence. **Right**: p(True) confidence where the token probability of "YES" (p) in **green** color or "NO" (1-p) is used as confidence.

# B ALGORITHM

---

**Algorithm 1:** Algorithm for LLM-ITL

---

**Input:** Train documents; An LLM; Pre-trained word embeddings; Hyper-parameters $T^{\text{refine}}$, $\gamma$; Training iteration $I$; Number of topics $K$.

**Initialize:** Initialize the parameters $\theta, \phi$ of the NTM.

  /\*Warm-up\*/
  **for** $i = 1 : T^{\text{refine}}$ **do**
    Compute NTM loss by Eq. 1;
    Compute gradients w.r.t $\theta$ and $\phi$;
    Update $\theta$ and $\phi$ based on the gradients;

**end**
/\*Topic Refinement\*/
  **for** $i = T^{\text{refine}} : I$ **do**
    **for** $k = 1 : K$ **do**
      Obtain topic distribution $t_k$ by Eq. 2;
      Obtain topic words $w_k$ by Eq. 3;
      Obtain refined words $w_k^{'}$ from the LLM by Eq. 5;
      Construct OT cost matrix by Eq. 7;
      Compute OT distance by Eq. 6;
      **if** *Open-Source LLM* **then**
        | Compute topic labeling confidence by Eq. 8
      **end**
      **else**
        | Compute topic labeling confidence by Eq. 9
      **end**
    **end**
    Compute $\mathcal{L}^{\text{refine}}$ by Eq. 10;
    Compute $\mathcal{L}^{\text{ntm}}$ by Eq. 1;
    Compute overall loss by $\mathcal{L}^{\text{ntm}} + \gamma \cdot \mathcal{L}^{\text{refine}}$ ;
    Compute gradients w.r.t $\theta$ and $\phi$;
    Update $\theta$ and $\phi$ based on the gradients;

**end**
**Output:** Trained NTM with $\theta, \phi$.

---

## C  RELATED WORK: LLM UNCERTAINTY ESTIMATION

Uncertainty estimation for LLMs (Geng et al., 2024) is emerging with the rapid usage of LLMs and their risk of hallucination (Ji et al., 2023). Sequence probability (Ren et al., 2022) leverages token probabilities to measure answer confidence. Verbalized confidence (Tian et al., 2023; Xiong et al., 2023) utilizes the LLM's own capability to evaluate its answer uncertainty. Consistency-based (Lin et al., 2023; Manakul et al., 2023) approaches sample multiple outputs from the LLM and measure answer consistency as uncertainty. Entropy-based (Kuhn et al., 2023; Hou et al., 2023) approaches use multiple LLM outputs to estimate the output space and compute entropy as uncertainty for the answer. Hybrid frameworks (Chen & Mueller, 2023; Gao et al., 2024) combine different approaches for a comprehensive estimation. Internal states (Chen et al., 2024) are another useful source for LLM uncertainty quantification. Unlike those works that estimate uncertainty for LLMs in general natural language generation tasks, ours focuses on task-specific uncertainty of the LLM in suggesting topic words.

## D  DETAILED EXPERIMENTAL SETTINGS

### D.1  DETAILS OF DATASET

Table D1: Statistics of the Datasets

| Dataset | # Docs Train | # Docs Test | Voc Size | Avg. Doc Length | # Labels |
|---------|--------------|-------------|----------|-----------------|----------|
| 20News  | 11778        | 2944        | 13925    | 150             | 20       |
| R8      | 5485         | 2189        | 5338     | 102             | 8        |
| DBpedia | 15598        | 3899        | 8550     | 51              | 14       |
| AGNews  | 16000        | 4000        | 8389     | 38              | 4        |

We conduct experiments on 20News [12], R8 [13], DBpedia [14], and AGNews [15]. For DBpedia and AG-News, we randomly sample a subset of 20,000 documents. We retain the original text documents for models that accept text as input, and preprocess the documents into Bag-of-Words (BOW) format for models that are trained on BOWs. We convert the documents into BOW vectors through the following steps: First, we clean the documents by removing special characters and stop words, followed by tokenization. Next, we build the vocabulary by including words with a document frequency greater than five and less than 80% of the total documents. Since we use the pre-trained word embeddings of GloVe (Pennington et al., 2014), we further filter the vocabulary by retaining only the words that are in the GloVe vocabulary. Finally, we transform the documents into BOWs based on the filtered vocabulary set. The statistics of the preprocessed datasets are summarized in Table D1.

### D.2  DETAILS OF BASELINES

We run the following topic models as our baselines, including Latent Dirichlet Allocation (**LDA**) (Blei et al., 2003), the most popular probabilistic topic model that generates documents by mixtures of topics; Neural Variational Document Model (**NVDM**) (Miao et al., 2017), a pioneering NTM based on the VAE framework; LDA with Products of Experts (**PLDA**) (Srivastava & Sutton, 2017), an NTM that uses a product of experts instead of the mixture model in LDA; Embedded Topic Model (**ETM**) (Dieng et al., 2020), which involves word and topic embeddings in the generative process of documents; Neural Topic Model with Covariates, Supervision, and Sparsity (**SCHOLAR**) (Card et al., 2017), an NTM that leverages extra information from metadata; Contrastive Learning Neural Topic Model (**CLNTM**) (Nguyen & Luu, 2021), an NTM that is based on the contrastive learning framework; **BERTopic** (Grootendorst, 2022), a recent clustering-based topic model that utilizes

---

[12]https://huggingface.co/datasets/SetFit/20_newsgroups

[13]https://huggingface.co/datasets/yangwang825/reuters-21578

[14]https://huggingface.co/datasets/fancyzhx/dbpedia_14

[15]https://huggingface.co/datasets/fancyzhx/ag_news

embeddings from pre-trained language models; **TopicGPT** (Pham et al., 2023), a latest LLM-based topic model that leverages an LLM for topic generation and assignment.

As for the implementations of baseline models, we use Mallet [16] for LDA with Gibbs sampling, and the original implementations for the other models. For NTMs including NVDM, PLDA, ETM, SCHOLAR and CLNTM, we tune their hyper-parameters for our datasets; For BERTopic, we fine-tune the topic representations after the topics are learned, as suggested by their implementation[17]. For TopicGPT, we use GPT-4 for topic generation and GPT-3.5 for topic assignment, randomly sampling 600 documents from the training set for each dataset, as suggested by their paper. We run all models except TopicGPT five times in each experiment and report the mean and standard deviation of their performance. For TopicGPT, we run it once for each experiment using a temperature value of zero to enable deterministic output, following the setting of their paper.

### D.3 DETAILS OF EVALUATION METRICS

For topic evaluation, we apply the commonly-used topic coherence metric, Normalized Pointwise Mutual Information (**NPMI**) (Lau et al., 2014), which evaluates topic coherence based on the co-occurrence of the topic's top words in a reference corpus. We use Wikipedia as the reference corpus for NPMI and consider the top 10 words of each topic, with implementation done using the Palmetto package[18] (Röder et al., 2015). We report the average NPMI score (in percentage) of all learned topics. For documents' topical representation (i.e., topic proportion) evaluation, a common practice is to compare the document clusters formed by topic proportions with those formed by the documents' true labels, known as topic alignment. Following previous works (Chuang et al., 2013; Pham et al., 2023), we assign each test document to a cluster based on the top-weighted topic of its topical representation, and compute Purity and Normalized Mutual Information (NMI) based on the documents' cluster assignments and their true labels. As Purity and NMI are often reported together and within the same range, we report the average score of both metrics as **PN**. For all evaluations, we use the model state at the end of the training iteration to compute all evaluation metrics.

---

[16]https://radimrehurek.com/gensim_3.8.3/models/wrappers/ldamallet.html

[17]https://maartengr.github.io/BERTopic/index.html

[18]https://github.com/dice-group/Palmetto

# E  MORE RESULTS

## E.1  TOPIC COHERENCE AT DIFFERENT K

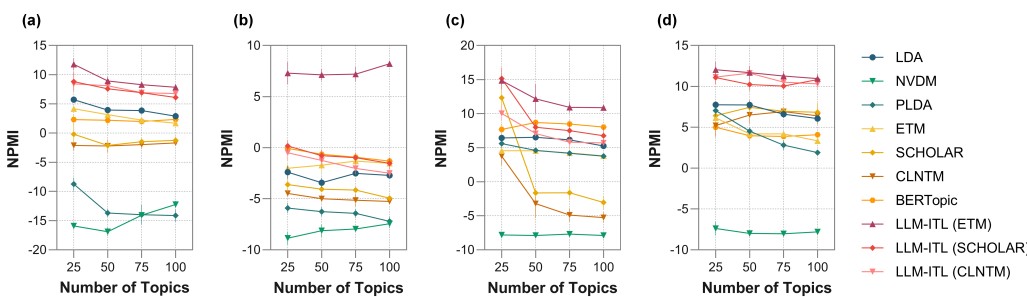

Figure E1: Topic coherence (NPMI) at different settings of the number of topics (i.e., $K$) on (**a**) 20News, (**b**) R8, (**c**) DBpedia and (**d**) AGNews.

Here, we illustrate the topic coherence performance of topic models across different numbers of topics (i.e., $K$). The results are shown in Figure E1. We observe that LLM-ITL consistently enhances topic coherence in its base models, and achieves state-of-the-art performance across various settings of $K$ in most cases.

## E.2  TOPIC ALIGNMENT AT DIFFERENT K

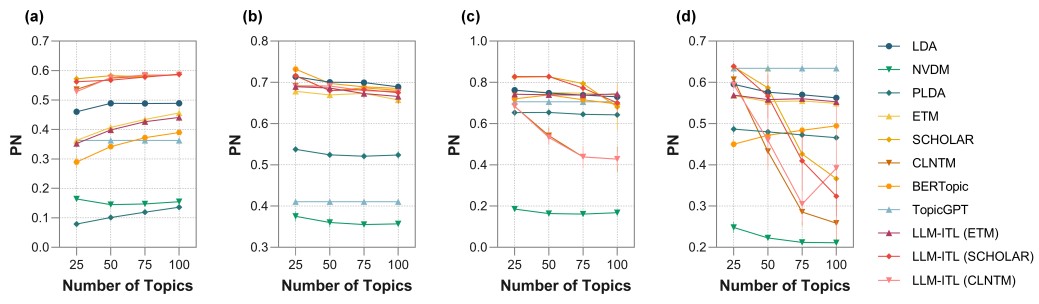

Figure E2: Topic alignment (PN) at different settings of the number of topics (i.e., $K$) on (**a**) 20News, (**b**) R8, (**c**) DBpedia and (**d**) AGNews.

Here, we illustrate the topic alignment performance of topic models across different numbers of topics (i.e., $K$). The results are shown in Figure E2. We observe that LLM-ITL consistently inherits the topic alignment performance of its base model across different settings of $K$. The performance drop with an increase in $K$ for SCHOLAR and CLNTM in short document datasets (e.g., DBpedia and AGNews) is due to their sensitivity to $K$ in short documents.

### E.3 TOPIC DIVERSITY AT DIFFERENT K

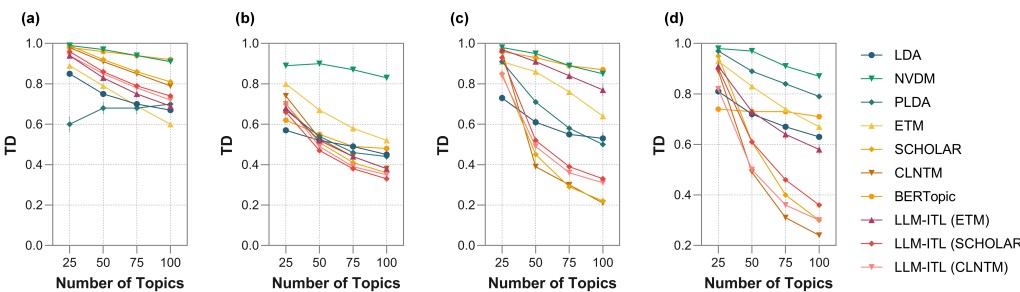

Figure E3: Topic diversity (TD) at different settings of the number of topics (i.e., $K$) on (**a**) 20News, (**b**) R8, (**c**) DBpedia and (**d**) AGNews.

Here, we illustrate the topic diversity (TD) performance of topic models across different numbers of topics (i.e., $K$). The results are presented in Figure E3. We observe that LLM-ITL demonstrates comparable performance in terms of topic diversity compared to its base models in most cases.

### E.4 PURITY & NMI

As we report PN (the mean of Purity and NMI) as an overall topic alignment metric in previous sections, we illustrate detailed Purity and NMI performance in this section. As shown in Table E1 and E2, Figure E4 and E5, LLM-ITL consistently inherits topic alignment performance from its base models in terms of both Purity and NMI, which is consistent with our previous observations based on PN.

Table E1: Purity performance. The best and second-best scores are highlighted in boldface and underlined, respectively. The performance improvement of LLM-ITL over its base model is computed.

| Model | 20News | R8 | DBpedia | AGNews |
|---|---|---|---|---|
| LDA | 0.521 ± 0.012 | 0.920 ± 0.004 | 0.813 ± 0.018 | 0.818 ± 0.010 |
| NVDM | 0.169 ± 0.007 | 0.603 ± 0.013 | 0.220 ± 0.004 | 0.724 ± 0.013 |
| PLDA | 0.130 ± 0.004 | 0.771 ± 0.011 | 0.730 ± 0.007 | 0.724 ± 0.013 |
| BERTopic | 0.371 ± 0.010 | 0.875 ± 0.002 | 0.748 ± 0.009 | 0.648 ± 0.014 |
| TopicGPT | 0.336 ± 0.000 | 0.577 ± 0.000 | 0.718 ± 0.000 | 0.819 ± 0.000 |
| ETM | 0.410 ± 0.011 | 0.875 ± 0.005 | 0.794 ± 0.016 | 0.784 ± 0.008 |
| LLM-ITL (ETM) | 0.403 ± 0.015 | 0.875 ± 0.012 | 0.761 ± 0.019 | 0.784 ± 0.006 |
| | ↓ 0.007 | ↑ 0.000 | ↓ 0.033 | ↑ 0.000 |
| SCHOLAR | **0.627** ± 0.014 | 0.911 ± 0.015 | 0.872 ± 0.018 | **0.855** ± 0.004 |
| LLM-ITL (SCHOLAR) | 0.607 ± 0.014 | 0.911 ± 0.015 | **0.875** ± 0.015 | **0.855** ± 0.004 |
| | ↓ 0.020 | ↑ 0.000 | ↑ 0.003 | ↑ 0.000 |
| CLNTM | 0.623 ± 0.015 | **0.923** ± 0.009 | 0.725 ± 0.047 | 0.821 ± 0.013 |
| LLM-ITL (CLNTM) | 0.623 ± 0.006 | 0.923 ± 0.008 | 0.727 ± 0.045 | 0.806 ± 0.014 |
| | ↑ 0.000 | ↑ 0.000 | ↑ 0.002 | ↓ 0.015 |

Table E2: NMI performance. The best and second-best scores are highlighted in boldface and underlined, respectively. The performance improvement of LLM-ITL over its base model is computed.

| Model | 20News | R8 | DBpedia | AGNews |
|---|---|---|---|---|
| LDA | 0.456 ± 0.006 | 0.481 ± 0.009 | 0.712 ± 0.008 | 0.373 ± 0.007 |
| NVDM | 0.120 ± 0.005 | 0.118 ± 0.011 | 0.151 ± 0.004 | 0.068 ± 0.007 |
| PLDA | 0.072 ± 0.005 | 0.277 ± 0.007 | 0.576 ± 0.009 | 0.250 ± 0.008 |
| BERTopic | 0.312 ± 0.007 | **0.519** ± 0.002 | 0.691 ± 0.008 | 0.252 ± 0.007 |
| TopicGPT | 0.390 ± 0.000 | 0.244 ± 0.000 | 0.694 ± 0.000 | **0.449** ± 0.000 |
| ETM | 0.405 ± 0.008 | 0.463 ± 0.014 | 0.728 ± 0.012 | 0.352 ± 0.007 |
| LLM-ITL (ETM) | 0.394 ± 0.005 | 0.498 ± 0.013 | 0.723 ± 0.014 | 0.354 ± 0.005 |
| | ↓ 0.011 | ↑ 0.035 | ↓ 0.005 | ↑ 0.002 |
| SCHOLAR | **0.538** ± 0.006 | 0.449 ± 0.012 | 0.778 ± 0.012 | 0.420 ± 0.006 |
| LLM-ITL (SCHOLAR) | 0.529 ± 0.006 | 0.449 ± 0.010 | **0.781** ± 0.012 | 0.423 ± 0.004 |
| | ↓ 0.009 | ↑ 0.000 | ↑ 0.003 | ↑ 0.003 |
| CLNTM | 0.526 ± 0.008 | 0.459 ± 0.004 | 0.641 ± 0.035 | 0.392 ± 0.016 |
| LLM-ITL (CLNTM) | 0.529 ± 0.005 | 0.459 ± 0.003 | 0.640 ± 0.034 | 0.382 ± 0.008 |
| | ↑ 0.003 | ↑ 0.000 | ↓ 0.001 | ↓ 0.010 |

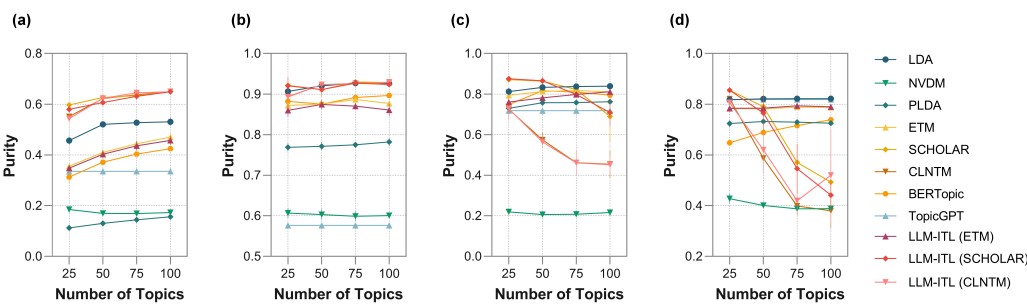

Figure E4: Purity performance at different settings of the number of topics (i.e., $K$) on (**a**) 20News, (**b**) R8, (**c**) DBpedia and (**d**) AGNews.

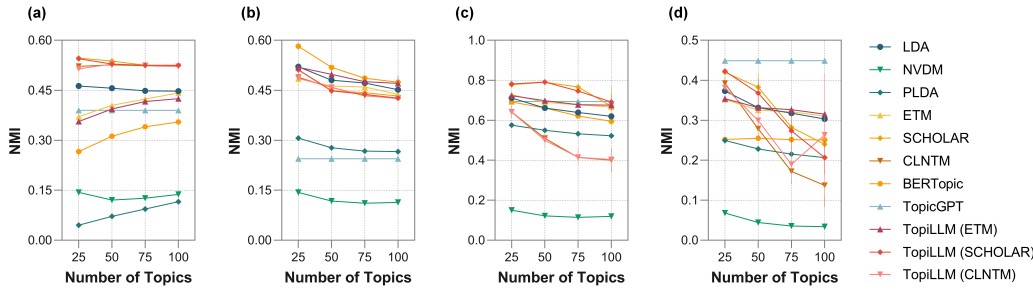

Figure E5: NMI performance at different settings of the number of topics (i.e., $K$) on (**a**) 20News, (**b**) R8, (**c**) DBpedia and (**d**) AGNews.

# F  CONFIDENCE ALTERNATIVES

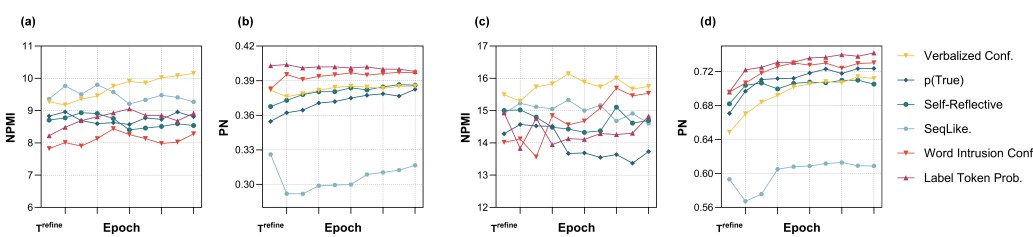

Figure F1: Study of confidence alternatives. Learning curves of LLM-ITL (ETM) with different confidence in terms of topic coherence (NPMI) and topic alignment (PN) on 20News ((**a**) and (**b**)) and DBpedia ((**c**) and (**d**)). Error bars are omitted for clarity in the figure.

Considering the efficiency of including confidence scores during the training of LLM-ITL, we focus solely on single-sample approaches for LLM uncertainty estimation in our study, where we run a single round of LLM inference for a given topic. We consider the following confidence alternatives to our topic labeling confidence for our study: (1) **No Conf**: No LLM confidence estimation is included, and $\mathrm{Conf}(\boldsymbol{w}_k^l) = 1$ in Eq. 10 during the training. (2) **Verbalized confidence** (Xiong et al., 2023), which directly asks the LLM for its confidence in solving a problem. The prompt we used for eliciting verbalized confidence is shown in Figure A2. (3) **Self-Reflective confidence** (Chen & Mueller, 2023), which prompts the LLM to evaluate its own answer in a two-stage manner. The topic label is obtained in the first stage, and the LLM evaluates this answer in a follow-up question (Figure A3). (4) **p(True)** (Kadavath et al., 2022), which is similar to self-reflective confidence, but asks a true/false question instead. It takes the token probability of the response as the confidence measure (Figure A3). (5) **SeqLike** (Ren et al., 2022), which computes the length-normalized sequence likelihood of the output from the LLM.

From the results in Figure F1, we can observe that while verbalized confidence improves topic coherence better, it biases the topics towards the LLM's knowledge rather than the topics within the corpus, leading to reduced topic alignment. On the other hand, label token probability and word intrusion confidence consistently yield the best topic alignment performance, suggesting greater relevance of the topics to the corpus.

# G  HYPER-PARAMETER STUDIES

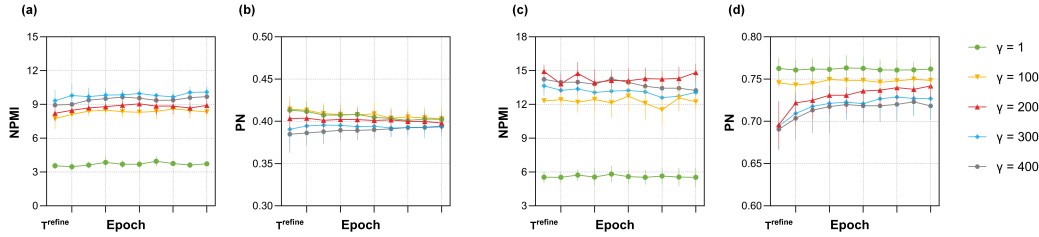

Figure G1: Learning curves of LLM-ITL (ETM) with different $\gamma$ in terms of NPMI and PN on 20News ((**a**) and (**b**)) and DBpedia ((**c**) and (**d**)).

Here, we study the hyper-parameter of LLM-ITL, focusing on topic refinement strength $\gamma$. We vary its value from 1 to 400 and plot the learning curves in terms of NPMI and PN, as shown in Figure G1. We can observe that: (1) In terms of topic coherence, $\gamma$ values between 100 and 300 yield similar performance, suggesting low sensitivity to $\gamma$ within this range. (2) In terms of topic alignment, a higher $\gamma$ leads to slightly reduced performance in the initial phase. This occurs because relying

heavily on topic refinement from the LLM causes the topics to bias towards the LLM's knowledge rather than the information from the input corpus. However, as training progresses, the performance converges to similar values. These observations suggest low sensitivity to $\gamma$ and its flexibility in controlling the balance between the information from the corpus and the knowledge from the LLM.

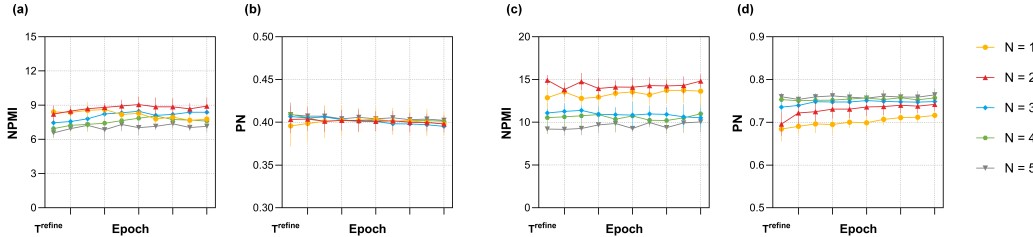

Figure G2: Learning curves of LLM-ITL (ETM) with different settings for the number of words (i.e., $N$) in the topic label, evaluated in terms of NPMI and PN on 20News ((**a**) and (**b**)) and DBpedia ((**c**) and (**d**)).

Here, we examine the hyper-parameter within the prompt, i.e., the number of words $N$ used for the topic label. We vary the number of words for the topic label from 1 to 5 and plot the learning curves in terms of NPMI and PN. As illustrated in Figure G2, we can observe that: (1) Using more words, such as a 5-word topic label (e.g., $N = 5$), results in the least improvement in topic coherence, while using a 2-word topic label (e.g., $N = 2$) achieves the best performance. (2) As for topic alignment performance, the number of words in the topic label shows comparable performance.

# H STUDY OF PROMPTS

Table H1: Prompt variants performance. The best and second-best performance of each column are highlighted in boldface and underlined, respectively.

| Prompt | Success Rate ($\uparrow$) | N_Input ($\downarrow$) | N_Output ($\downarrow$) | Refined TC ($\uparrow$) |
|---|---|---|---|---|
| Origin | 0.978 | 224.48 | 197.86 | 7.165 |
| Variant_1 | 0.967 | 228.48 | **80.81** | 4.520 |
| Variant_2 | 0.935 | 269.48 | 167.05 | 3.720 |
| Variant_3 | 0.980 | 223.48 | 172.51 | 6.146 |
| Variant_4 | 0.972 | **189.48** | 191.04 | **7.490** |
| Variant_5 | 0.956 | 208.48 | 159.66 | 6.215 |
| Iterative Refinement (Chang et al., 2024) | **0.993** | 1872.13 | 936.70 | 2.845 |

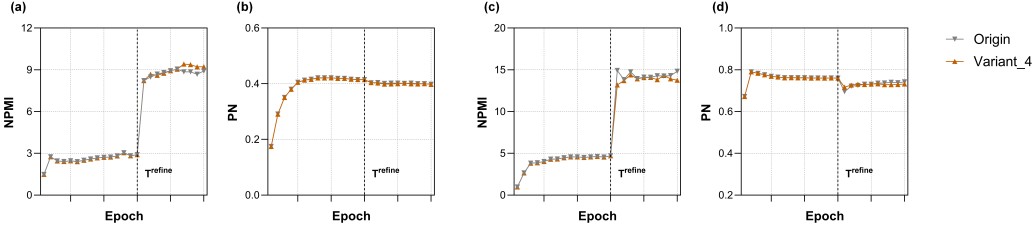

Figure H1: Learning curves of LLM-ITL (ETM) with different prompts in terms of NPMI and PN on 20News (figure (**a**) and (**b**)) and DBpedia (figure (**c**) and (**d**)).

**Prompt Variants**   Here, we study the effectiveness of different prompts for topic suggestion. We obtain variants of topic suggestion prompts in Figure A1 through modifying the technique used in PromptBreeder (Fernando et al., 2023). To be specific, we first create a set of 100 'mutation-prompts' (e.g., "Make a variant of the prompt") and 100 'thinking-styles' (e.g., "Let's think step by step"). We generate a set of 50 task prompts by concatenating a randomly drawn 'mutation-prompt' and a randomly drawn 'thinking-style' to the original prompt, and provide that to the Claude 3.5[19] to produce a continuation, resulting in a different task prompt. Secondly, we randomly select 50 topics over 4 experiment datasets. We run those topics through 50 generated task prompts and filter out the generated prompts that cannot give JSON format in the selected topics or generate above 300 tokens. We are left with 14 topics. We then leverage Claude 3.5 to judge the quality of generated topics and refined topic words. We rank 14 methods by overall topics and refined topic words to get 5 variants of prompts. In addition to the prompt variants generated by those steps, we also investigate the topic refinement prompt used in Chang et al. (2024) (see Figure 2 of their paper). All the prompt variants for topic refinement in this study are illustrated in Table H2.

**Setup**   We randomly sample 1000 topics learned by topic models, then use different prompts to refine the topics with LLAMA3-8B-Instruct[20]. We analyze the effectiveness of prompts in different aspects, including **Success Rate** (Ulmer et al., 2024): the proportion of cases where the target answer can be successfully extracted from the LLM's output; **N_Input** and **N_Output** (Chang et al., 2024): the average number of tokens of input and output of the LLM; and **Refined TC**: the average NPMI scores of the refined topics.

**Results**   From the results in Table H1, we observe the following: (1) Through prompt optimization, the effectiveness of the prompt can be further enhanced (e.g., Variant_4), where the number of tokens (i.e., the cost) is reduced and the refine topics are more coherent. (2) The iterative refinement (Chang et al., 2024) shows less effectiveness in terms of both cost and refined topic coherence compared with our prompt variants when applied to LLAMA3-8B-Instruct.

Based on the above observations, we further investigate the effectiveness of the improved prompt within the LLM-ITL framework. We plot the learning curves of LLM-ITL using the original prompt and its variant (Variant_4, which shows better performance from Table H1). We observe that the overall performance in terms of both metrics is comparable.

---

[19]https://www.anthropic.com/news/claude-3-family
[20]https://huggingface.co/meta-llama/Meta-Llama-3-8B-Instruct

Table H2: Prompt variants for topic refinement

|  | Prompt |
|---|---|
| Origin | Analyze step-by-step and provide the final answer.
Step 1. Given a set of words, summarize a topic (avoid using proper nouns as topics) by 2 words that covers most of those words. Note, only the topic, no other explanations.
Step 2. Remove irrelevant words about the topic from the given word list. Note, only the removed words, no other explanations.
Step 3. Add new relevant words (maximum 10 words) about the topic to the word list up to 10 words. Note, only the added words, no other explanations.
Step 4. Provide your answer in json format as {'Topic': '<2 Word Topic>', 'Words': '<Refined 10 Word List>'}. Note, only 10 refined words allowed for the topic, and no follow up explanations. |
| Variant_1 | Perform the following actions sequentially and provide the final result:
Step 1. After examining a set of words, condense a subject (avoid proper nouns) into 2 words that encompass most of those words. (Note: Only the subject, no further elaboration.)
Step 2. Eliminate irrelevant words from the given word list based on the subject. (Note: Only the removed words, no further elaboration.)
Step 3. Add new pertinent words (maximum 10 words) related to the subject to the word list until it reaches 10 words. (Note: Only the added words, no further elaboration.)
Step 4. Present your response in JSON format as {'Topic': '<2 Word Subject>', 'Words': '<Refined 10 Word List>'}. Note: Only 10 refined words are permitted for the subject, and no follow-up explanations. |
| Variant_2 | Perform a meticulous examination and furnish the conclusive resolution.
Stride 1. Bestowed a catalogue of vocabularies, condense a subject matter (circumvent the employment of proper appellations as subjects) by dual words that envelop the preponderance of those vocabularies. (Heed, solely the subject, devoid of supplemental explication.)
Stride 2. Dislodge irrelevant vocabularies concerning the subject from the granted vocabulary catalogue. (Heed, solely the dislodged vocabularies, devoid of supplemental explication.)
Stride 3. Amalgamate novel applicable vocabularies (maximal 10 vocabularies) concerning the subject to the vocabulary catalogue up to 10 vocabularies. (Heed, solely the amalgamated vocabularies, devoid of supplemental explication.)
Stride 4. Tender your resolution in json format as {'Topic': '<2 Word Subject>', 'Words': '<Refined 10 Word Catalogue>'}. Heed, solely 10 refined vocabularies permitted for the subject, and devoid of successive explication. |
| Variant_3 | Step-by-step analysis and final answer:
Step 1. Given a set of words, summarize a topic (avoid using proper nouns as topics) by 2 words that covers most of those words. (Note, only the topic, no other explanations.)
Step 2. Remove irrelevant words about the topic from the given word list. (Note, only the removed words, no other explanations.)
Step 3. Add new relevant words (maximum 10 words) about the topic to the word list, keeping the total word count at 10 words. (Note, only the added words, no other explanations.)
Step 4. Provide your answer in JSON format as {'Topic': '<2 Word Topic>', 'Words': '<Refined 10 Word List>'}. Note, only 10 refined words allowed for the topic, and no follow-up explanations. |
| Variant_4 | Break down the analysis into steps and give the final response.
1. Look at a set of words and identify a 2-word topic that sums up most of those words (don't use proper nouns as topics, just state the topic).
2. Remove words from the list that don't relate to the topic (just list the removed words).
3. Add new relevant words about the topic to the list, up to 10 words total (just list the new added words).
4. Provide your response in JSON format: {'Topic': '<2 Word Topic>', 'Words': '<Refined 10 Word List>'}. Only include 10 words for the refined list, no explanations. |
| Variant_5 | Step-by-step analysis and provide the final answer in JSON format:
Step 1: Based on the given set of words, summarize a topic using 2 words that encompass most of those words (avoid proper nouns).
Step 2: Remove any irrelevant words from the given word list that do not relate to the summarized topic.
Step 3: Add new relevant words (up to 10 words) that are related to the summarized topic.
Step 4: Present your answer in the following JSON format: {'Topic': '<2 Word Topic>', 'Words': '<Refined 10 Word List>'}, where 'Topic' contains the 2-word summarized topic, and 'Words' contains the refined list of 10 words related to that topic. Do not provide any additional explanations. |
| Iterative Refinement (Chang et al., 2024) | Please analyze the following tasks and provide your answer in the specified format.
1. Determine the common topic shared by these words: [<TOPIC_WORDS>].
2. Assess whether the word "<WORD>" aligns with the same common topic as the words listed above. Respond with:
- "Yes", if the given word shares the common topic.
- If "No", suggest 10 single-word alternatives that are commonly used and closely related to this topic. These words should be easily recognizable and distinct from the ones in the provided list.
Format your response in JSON, including the fields "Topic", "Answer", and "Alternative words" (only if the answer is "No"). |

