# OpenReview forum: "Neural Topic Modeling with Large Language Models in the Loop"
_ICLR.cc/2025/Conference — ICLR 2025 Conference Withdrawn Submission_

### Official Review · Reviewer_biVA · 2024-10-29

**Soundness:** 2
**Presentation:** 3
**Contribution:** 2
**Rating:** 3
**Confidence:** 5

**Summary:**

This paper proposes LLM-ITL, a new LLM-in-the-loop framework that integrates LLMs with existing Neural Topic Models (NTMs). In LLM-ITL, global topics and document representations are learned through the NTM, while an LLM refines the topics via a confidence-weighted Optimal Transport (OT)-based alignment objective. Unlike most existing LLM-based approaches that rely on document-level LLM analysis, the proposed LLM-ITL uses LLMs at the word level, which is novel to me. However, many important and strong existing baselines are missing in this work, rendering the central claims of the paper inadequately supported by evidence.

**Strengths:**

- The presentation is generally easy to follow and the content is concise.

- Unlike most existing LLM-based approaches that rely on document-level LLM analysis, the proposed LLM-ITL uses LLMs at the word level, which is novel to me.

**Weaknesses:**

- In the area of neural topic modeling, an important and strong existing model is ECRTM (Wu et al., 2023), but this submission overlooks the above study. The other strong baseline is WeTe (Wang et al., 2022), and I suggest that the authors introduce ECRTM and WeTe as baselines. Besides, as mentioned in (Wu et al., 2023), the coherence value of $C_V$ has been empirically shown to outperform the traditional metrics such as NPMI. Thus, I suggest that the authors improve their descriptions on the selection of evaluation metrics.

- NSTM (Zhao et al., 2020) learns the topic proportions by minimizing the OT distance to the document-word distribution. Although this submission has cited NSTM, it does not employ NSTM as a baseline for comparison. Considering that both works introduce the OT distance, I think NSTM should be compared in the experiments.

References:
(Wang et al., 2022) D. Wang, et al. Representing mixtures of word embeddings with mixtures of topic embeddings. ICLR 2022.
(Wu et al., 2023) X. Wu, et al. Effective neural topic modeling with embedding clustering regularization. ICML 2023.
(Zhao et al., 2020) H. Zhao, et al. Neural topic model via optimal transport. arXiv preprint arXiv:2008.13537, 2020. Note that this work has been published at ICLR 2021.

**Questions:**

- In Step 1 of Figure A1, why you specify avoiding using proper nouns as topics and summarizing a topic with a fixed value of 2 words?

- As you cited from (Doi et al., 2024), LLMs are asked to focus on a document individually; they may be unable to cover all the topics across all the documents in the corpus. Besides, with their limited window of focus, LLMs may miss key topics of a document that are necessary to fully understand its content. Given that the results of LLMs on topic modeling are biased, why do you trust the output of LLMs and try to align the topic words generated by neural topic models with those suggest by LLMs?

---

### Official Review · Reviewer_SmUe · 2024-11-03

**Soundness:** 2
**Presentation:** 3
**Contribution:** 2
**Rating:** 3
**Confidence:** 4

**Summary:**

This paper aims to design a topic model with the aid of LLMs and optimal transport to seek the balance between topic quality and document representations. Specifically, this paper introduces an LLM-based refinement step, which is guided by optimal transport. A confidence weighted mechanism is presented to alleviate the hallucination effect of LLMs. Experiments on topic coherence, topic diversity, topic alignment, and document clustering verify the effectiveness of the proposed model.

**Strengths:**

1. With the modeling power of LLMs, their applications on topic modeling are also emerging. This paper proposes to use LLMs to help topic modeling, which is timingly significant.

2. The overall writing of the paper is clear with figures as visual illustration and a formal algorithm to present the learning process. Details of experimental setup are also provided.

3. Experiments are comprehensive with different evaluation tasks. Ablation analysis is also conducted to show the effect of each modeling component.

**Weaknesses:**

Though this paper proposes an interesting approach, there are some unacceptable shortcomings.

1. One of the contribuitons mentioned in the Introduction section is scalability, which is evaluated by the number of parameters and running time in the Computational Costs section. However, when we talk about scalability, both empirical and theoretical analyses are important, but there is a lack of computational complexity analysis in the paper, which limits the contribution proposed in the paper.

2. An important evaluation task for topic modeling is document completion, which is evaluated by perplexity. This is a widely adopted task, including LDA (Blei et al., 2003) and ProdLDA (Srivastava & Sutton, 2017). There is a lack of perplexity experiment in the paper.

3. The datasets used for experiments are relatively small, considering that one of the contributions in the paper is scalability. I am expecting authors to test on million-sized datasets to clearly show that the proposed model is scalable on large-scale datasets.

4. For some experiments, such as clustering with Purity and NMI, the proposed model doesn't outperform baselines, such as in Figure E4, which limits the effectiveness of the proposed model. There is also a lack of explanation on why the proposed model doesn't outperform baselines.

5. Using optimal transport for topic modeling has been a while, and there are some papers working on this problem. The submitted paper misses to mention some highly related papers [1-3].

[1] Zhang, D. C., & Lauw, H. W. (2023). Topic Modeling on Document Networks with Dirichlet Optimal Transport Barycenter. IEEE Transactions on Knowledge and Data Engineering.

[2] Huynh, V., Zhao, H., & Phung, D. (2020). Otlda: A geometry-aware optimal transport approach for topic modeling. Advances in Neural Information Processing Systems, 33, 18573-18582.

[3] Xu, H., Wang, W., Liu, W., & Carin, L. (2018). Distilled wasserstein learning for word embedding and topic modeling. Advances in Neural Information Processing Systems, 31.

**Questions:**

1. Why is standard deviation not reported for most figures in the paper?

---

### Official Review · Reviewer_4R8J · 2024-11-04

**Soundness:** 1
**Presentation:** 3
**Contribution:** 2
**Rating:** 3
**Confidence:** 5

**Summary:**

This paper proposes LLM-ITL, an LLM-in-the-loop topic modeling framework. The approach refines topics learned by existing NTMs using LLMs. At a defined refinement step, an LLM is prompted to suggest topics based on those learned by the NTM. Optimal Transport is then used to align the NTM topics with the LLM-suggested topics, with alignment weighted by the LLM's confidence in its topic suggestions. This refinement objective is incorporated into the standard topic modeling objective as the overall objective of LLM-ITL.

**Strengths:**

1. Incorporating LLMs in topic modeling is a relatively unexplored, challenging, and important direction.
2. The paper is easy to read and understand. The authors have done a good job to present their work.

**Weaknesses:**

1. **Method:** Although this is a promising direction, I believe there are significant shortcomings with the method. From what I understand, LLMs can suggest out-of-vocabulary words, which means there is a risk of generating topics that do not represent the corpus at all and are simply highly coherent due to the LLM. I think this should be evaluated. One major issue is that the authors have used Wikipedia as a reference corpus to compute NPMI (which is generally acceptable), but in this case, NPMI should also be reported with the training set as the reference corpus. This would help gauge whether LLMs are refining topics with external information not present in the training set.

2. **Evaluation:** Why is NPMI reported as a percentage? Numerous studies have already highlighted that evaluating topic models is flawed and not standardized (see [1]). It would be better to report standardized coherence metrics, such as $C_{NPMI}$ or $C_V$ from [2]. These metrics have also been implemented in popular Python packages like Gensim.

3. **Performance:** In L1101-1102, it is stated that LLM-ITL demonstrates comparable performance with the baselines. I find this claim misleading, as Figure E3 indicates otherwise. There appears to be a considerable difference in the diversity between LLM-ITL and the best-performing baselines. Calculating the topic quality scores (TC x TD) would likely reveal that LLM-ITL does not perform as well as the baselines.

4. **Experiments:** Relatively outdated baselines are used overall. I do not believe that only using SCHOLAR, NVDM, and PLDA in 2024 is appropriate. Newer and extensively validated models, such as DVAE [1], NSTM, WeTe, and ECRTM [3], are available and should be included as baselines.

[1] Hoyle, Alexander, et al. "Is automated topic model evaluation broken? the incoherence of coherence." Advances in neural information processing systems 34 (2021): 2018-2033.
[2] Röder, Michael, Andreas Both, and Alexander Hinneburg. "Exploring the space of topic coherence measures." Proceedings of the eighth ACM international conference on Web search and data mining. 2015.
[3] Wu, X., Dong, X., Nguyen, T. T., & Luu, A. T. (2023, July). Effective neural topic modeling with embedding clustering regularization. In International Conference on Machine Learning (pp. 37335-37357). PMLR.

**Questions:**

1. How is the probability vector $\textit{u}$ constructed? How do you ensure that LLM is going to suggest (refined) words that are from the same vocabulary (V)?

---

### Official Review · Reviewer_obQ5 · 2024-11-06

**Soundness:** 2
**Presentation:** 3
**Contribution:** 2
**Rating:** 5
**Confidence:** 5

**Summary:**

This work addresses key challenges in enhancing Neural Topic Models (NTMs), specifically in improving topic quality/coherence and document representation by incorporating external knowledge. Leveraging a large language model (LLM) for topic refinement, the proposed approach first generates topics using a VAE-based NTM, then refines these topics with the LLM. This refinement is guided by a comparison of probability distributions via Optimal Transport, resulting in an augmented loss function for the NTM that includes a refinement term. The approach further handles LLM hallucinations into NTM training. This method is named as LLM-ITL.

To validate this approach, experiments were conducted on four datasets using two evaluation metrics. The results demonstrate improved topic coherence, measured by NMPI, and enhanced document representation quality through topic alignment.

**Strengths:**

- The paper proposes an approach to enhance topic quality through topic refinement, leveraging the capabilities of LLMs.
- While the approach represents progress in utilizing LLMs, it lacks substantial originality and novelty.
- The paper is clear and well-written; however, certain claims—such as improvements in document representation quality—are not clearly supported by the architecture or methodology.
- The experimental section is robust, utilizing four datasets and two evaluation metrics to validate the approach.
- The qualitative evaluation is thorough, including an ablation study for additional insights.

**Weaknesses:**

- The paper takes an incremental approach, building on prior work to enhance topic models using external knowledge or language models. Specifically, it extends Neural Topic Models (NTMs) through a topic refinement approach leveraging large language models (LLMs).
- The paper offers limited novelty in advancing the field, as previous studies have already improved baseline NTMs for topic modeling, extraction, and document representation.
- The paper claims to improve "document representation quality," but it is unclear from the architecture or methodology how this is achieved, given that the loss function primarily optimizes for topic coherence. Enhanced topic coherence does not always translate to better document representations.
- It is unclear how potential suboptimal quality in the LLM might impact NTM training outcomes.
- The method for refining topics is unclear, especially if the initial NTM output is incoherent. How does the LLM generate accurate topic labels and perform refinement in such cases?
- Baseline neural topic models such as DocNADE [1, 2] and references to foundational work [3, 4, 5] in this line of research, are absent from the experimental evaluations or comparative analysis.
- The paper lacks a key evaluation metrics that directly assess document representation quality, such as document retrieval (e.g., [2, 4]). This will further validate the applicability in real-world usecases.

Missing References (line 37, 46, 50, 281, 292):
[1] DocNADE. The neural autoregressive distribution estimator. AISTATS 2011.
[2] Document informed neural autoregressive topic models with distributional prior. AAAI 2019.
[3] TopicBERT for Energy Efficient Document Classification. EMNLP 2020.
[4] textTOvec: Deep Contextualized Neural Autoregressive Topic Models of Language with Distributed Compositional Prior. ICLR 2019.
[5] Topically driven neural language model. ACL 2017.

**Questions:**

- How does an LLM generate a new topic distribution when given NTM topics as input (represented by distributions)? How does it produce refined topics using only topic labels, without relying on document context?
- What motivates the use of the Optimal Transport (OT) metric for comparing probability distributions, rather than alternatives like the KL-divergence?
- In NTMs, Variational Autoencoders (VAEs) are typically used as the backbone architecture. However, models like DocNADE have shown better performance than VAEs, especially for document-topic representations. Given that OT alignment acts as a form of regularization in NTM training, how might this benefit other neural topic models like DocNADE?

---

### Note · Authors · 2024-11-13

I have read and agree with the venue's withdrawal policy on behalf of myself and my co-authors.